# Iron Plaque: A Shield against Soil Contamination and Key to Sustainable Agriculture

**DOI:** 10.3390/plants13111476

**Published:** 2024-05-27

**Authors:** Zeping He, Jinyuan Chen, Shilin Yuan, Sha Chen, Yuanyi Hu, Yi Zheng, Ding Li

**Affiliations:** 1School of Life Sciences and Chemistry, Hunan University of Technology, Zhuzhou 412007, China; hzping334748@163.com (Z.H.); slyuan2003@163.com (S.Y.); chensha@hut.edu.cn (S.C.); z1277716629@outlook.com (Y.Z.); 2Hunan Provincial Engineering Research Center of Lily Germplasm Resource Innovation and Deep Processing, Hunan University of Technology, Zhuzhou 412007, China; 3Zhuzhou City Joint Laboratory of Environmental Microbiology and Plant Resources Utilization, Hunan University of Technology, Zhuzhou 412007, China; 4State Key Laboratory of Hybrid Rice, Hunan Hybrid Rice Research Center, Changsha 410125, China; huyuanyi237@163.com; 5National Center of Technology Innovation for Saline-Alkali Tolerant Rice in Sanya, Sanya 572000, China

**Keywords:** iron plaque, hydrophytes, heavy metals

## Abstract

**Highlights:**

What are the main topic discussed?
Description of the formation process of iron plaque.The factors affecting the formation of iron plaque were summarized.

What are the implications of these discussions?
Understanding the role of iron plaque in environmental processes.Insights into the interactions between iron plaque, plants, and microbes for environmental remediation.

**Abstract:**

Soils play a dominant role in supporting the survival and growth of crops and they are also extremely important for human health and food safety. At present, the contamination of soil by heavy metals remains a globally concerning environmental issue that needs to be resolved. In the environment, iron plaque, naturally occurring on the root surface of wetland plants, is found to be equipped with an excellent ability at blocking the migration of heavy metals from soils to plants, which can be further developed as an environmentally friendly strategy for soil remediation to ensure food security. Because of its large surface-to-volume porous structure, iron plaque exhibits high binding affinity to heavy metals. Moreover, iron plaque can be seen as a reservoir to store nutrients to support the growth of plants. In this review, the formation process of iron plaque, the ecological role that iron plaque plays in the environment and the interaction between iron plaque, plants and microbes, are summarized.

## 1. Introduction

In the past few decades, the contamination of soils by heavy metals (HMs) has raised worldwide concerns due to the intensive human activities on the environment [1,2,3,4]. For instance, soils from the central area of Yueliangbao gold tailings (located in central China), were found to be rich in Cu, Pb, Zn, Mn, Mo and Cd, of which concentrations were much higher than that of these species in soils from the surrounding regions [5]. A total of 12 metal pollutants, including As, Cr and Hg, etc., were detected in the sediments of the Fuyang river system in north China, potentially risking local ecological safety and human health based on analyses of the geo-accumulation index and Pearson’s correlation [6]. In Florida (the United States), As and Pb concentrations in urban soils exceeded the local criteria for residential site soils by 2.1 mg kg^−1^ and 400 mg kg^−1^, respectively [7]. In the suburbs of Multan, a city in east-central Pakistan, the contents of Cd, Cu, Mn, Ni and Pb in *Brassica rapa,* which is commonly used as fodder for local animals, far exceeded the permissible limits that the World Health Organization (WHO) prescribed for *B. rapa* as animal fodder, resulting in high carcinogenic health risks to animals, as evidenced by the super high values of the total target health quotient (TTHQ), which ranged from 47.22 to 136.64 (TTHQ > 1 is an indicator of carcinogenic food stuffs, according to the US Environmental Protection Agency) [8].

Unlike organic pollutants that generally consist of carbon chains, HMs cannot be decomposed or eliminated in soils via chemical and biological processes [9]. Thus, they will be likely absorbed by plant roots from the soil. In plant tissues, an over-accumulation of HMs will interfere with various metabolic processes, such as damaging the protein structure, replacing essential metals in the biomolecules (e.g., pigments and enzymes), retarding cell division, inactivating photosynthesis and respiration, thereby resulting in significant inhibition of growth and loss of yields [10,11,12,13,14,15]. Accordingly, to safeguard against toxic HMs, plants have developed several elaborate strategies in vivo, including compartmentalization of HMs in cell organelles, inactivation of HMs by chelation with organic ligands, exclusion of HMs by using specific transporters and ion channels [16].

Apart from the above-mentioned in vivo strategies, another in vitro one, i.e., IP, which naturally occurs on the surface of wetland plant roots, was found to be effective in blocking the uptake of HMs by roots from soils [17,18,19,20,21]. In fact, IP can be considered as a respiratory by-product of plant roots grown in the submerged soils [22]. In this case, underground roots are in water-logged conditions that usually lack gaseous oxygen. Through well-developed aerenchyma, abundant oxygen is transferred from the overground tissues to the roots, mostly acting as electron receptors in root cells. Meanwhile, some oxygen and reactive oxygen species (ROS) that are generally produced along the respiratory chains may interact with various Fe species from the soil to generate IP, gradually enveloping the surface of roots. Because of the large surface-to-volume porous structure, IP often exhibits a high binding affinity to metal ions and hence can act as a sink for HMs and physically insulate HMs from the surface of roots [23].

To date, it is widely acknowledged that IP plays a pivotal role in safeguarding hydrophytes against HM toxicity. Therefore, comprehending the formation process and environmental functions of IP holds great significance for soil remediation. This review focuses on elucidating the mechanism underlying IP formation and its impact on impeding HM uptake by hydrophytes. Additionally, we discuss the effects of IP on soil properties and plant growth as well as the interactions between plants, IP and microbes.

## 2. Characteristics of IP

### 2.1. Discovery of IP

The discovery of IP could be traced back to as early as the 1960s. Armstrong (1967) [24] found that in two plant species (*Menyanthes trifoliata* and *Molinia coerulea*), the root oxidizing activity was the highest at the root apex, and gradually diminished towards the root base. Concomitantly, iron oxide deposits were formed to substantially accumulate around the root apical region, and once more, to gradually diminish towards the root base. However, it is worth noting that the apical region itself was commonly free of iron oxides because this region exhibited the greatest oxidizing activity, causing the oxidation of ferrous iron to occur at some distance from the root tip [24]. Later, Bacha and Hossner (1977) [25] demonstrated a positive correlation between the contents of iron precipitates formed on the roots of rice plants (*Oryza sativa* ‘Brazos’) and the initial concentrations of ferrous chloride added in soils. Moreover, they used the techniques of scanning electron microscopy (SEM) and X-ray diffraction (XRD) to examine the morphology and mineral structure of iron precipitates on the rice roots, showing that these iron precipitates corresponded to the poorly crystalline lepidocrocite (γ-FeOOH). By far, numerous studies have shown that IP can be considered as a natural armor to protect wetland plants (e.g., rice, reed and Typha, etc.) from HMs (Table 1), whilst also benefiting from adsorbing nutrients from soils [26,27,28,29,30]. Furthermore, IP can help rush plants survive in strongly acidic soils, where the concentration of inorganic carbon is commonly lower than that of natural soils [27]. Specifically speaking, it acts as a carbon sink to fix organic compounds exudated by rush roots, thereby allowing the rapid bacterial recycling of carbon elements back to plants. As a result, IP enabled rush roots to have access to relatively high concentrations of carbon sources that are required for growth metabolism (e.g., photosynthesis) in low-carbon soils [27].

### 2.2. Mineral Composition of IP

Chen et al. (1980) [46] proposed that when the oxidation of Fe(II) by O_2_ occurred, the formed iron oxide, FeOOH, was incipiently precipitated on the epidermal cell wall of rice roots. As the outermost cell wall decomposed, the FeOOH particles began to fill the cellular spaces to generate polyhedral casts [47]. Now, it is generally accepted that IP can be separated into two classes, i.e., amorphous and poorly crystalline IP, and the changes in environmental conditions such as redox potential (Eh), pH and Fe(II) concentration may favor the transformation between them. For instance, the crystallinity of IP that occurred on the roots of *Spartina alterniflora* increased with the Fe(II) concentration in the soil [48]. Crystalline IP mainly consists of iron oxides [32,49,50]. In the natural environment, there are at least 16 iron oxide counterparts (Table 2) but in most cases, only ferrihydrite (Fe_2_O_3_·nH_2_O), goethite [α-FeO(OH)] and lepidocrocite [γ-FeO(OH)] are believed to be the major components of crystalline IP [32], while sometimes, minor amounts of siderite (FeCO_3_) are also present in it [31].

The reddish-brown ferrihydrites, regardless of the natural and synthetic ones, are poorly crystalline iron oxide-hydroxides [49]. According to XRD patterns (the number of peaks in XRD spectra), ferrihydrites can be basically classified into five types: 2-line, 3-line, 4-line, 5-line and 6-line one, among which the 2-line and 6-line are seen as the two extremes of the crystal order for ferrihydrites, and are more prevalent than others in the environment [49]. As structural crystallinity increases, the 2-line ferrihydrite shows two reflections while the 6-line one displays six–eight reflections in the XRD spectra. Environmental reaction conditions play a significant role in shaping the crystallinity form of ferrihydrites. For instance, the crystalline lattice order decreases as the rate of Fe(III) hydrolysis increases, and as the concentration of silicate or soil organic anions increases [51].

The yellow-brown goethite, occurring throughout the global ecosystem, is one of the most thermodynamically stable iron oxides at environmental temperatures. Structurally, goethite is characterized by double chains of Fe octahedra, which are formed by edge-sharing and oriented parallel to the crystallographic direction. Within each octahedral unit, the Fe(III) cation is octahedrally coordinated by three O^2-^ anions and three OH- anions. Notably, the orthorhombic symmetry of goethite arises from the alternating arrangement of these double chains of Fe octahedra with double chains of vacant lattice sites [52]. In the idealized structure of goethite, the bond lengths between the Fe(III) cation and the surrounding oxygen atoms, denoted as d(Fe-O), exhibit two distinct values: 1.95 Å for three oxygen atoms and 2.09 Å for the remaining three oxygen atoms [53,54]. Moreover, the distinctive edge-sharing and double chain arrangement of the Fe octahedra in goethite give rise to three unique Fe-Fe distances, d(Fe-Fe), specifically 3.01 Å for two edge-sharing Fe atoms, 3.28 Å for another set of two edge-sharing Fe atoms and 3.46 Å for four Fe atoms sharing double corners [55].

Lepidocrocite (γ-FeO(OH)) is a naturally occurring iron oxide mineral widely distributed in the environment, playing a crucial role in the geochemical cycling of iron elements. The structural characteristic of lepidocrocite is the presence of double chains composed of FeO_6_ octahedra that share edges, parallel to the crystallographic *c*-axis. These chains are held together by hydrogen bonding between the oxygen atoms of the octahedral units. In each octahedral unit, the Fe(Ⅲ) cation is coordinated by five O^2−^ anions and one OH^-^ anion, with varying Fe-O bond lengths which reflect the different bonding environments of the oxygen atoms [49]. Lepidocrocite is known for its ability to adsorb various cations and anions from water, making it useful in natural water purification [56]. Additionally, lepidocrocite often coexists with goethite and other iron oxides in nature, also playing a critical role in the cycling of trace elements except for iron [57].

It should be noted that different types of iron oxides coexist in the natural synthesis process of IP, and they interact and transform with each other, ultimately forming a reddish-brown film on the roots of plants.

### 2.3. Formation of IP

Iron exists as an abundant transition metal element in the environment. As shown in Figure 1, in the early stage, after redox by root exudates and microbes, the iron elements (including different valence states) in the rhizosphere are transformed into soluble Fe(II). Then, O_2_ is supplied by radical oxygen loss (ROL) through the aeration tissues (aerenchyma), creating an oxygen-rich zone. In the middle and late stages, through the equation of 4Fe(II) + 10H_2_O + O_2_ → 4Fe(OH)_3_ + 8H^+^ [58,59,60], Fe(II) is transformed into iron oxide, which precipitates on the surface of the roots, thus forming IP and protecting plant roots from HMs via adsorption.

Previous studies have shown that IP is more likely to occur in acidic pH environments [61,62]. After a series of redox reactions, it ultimately precipitates on the root surface in the form of Fe_2_O_3_ or Fe(OH)_3_ [20]. Interestingly, the distribution of IP on the root surface is not uniform. IP is principally found in the elongation and root-hair zones of plant roots but is rarely observed in young lateral roots or newly formed roots [26,63]. This may result from the continuous growth of roots during plaque formation, where the older sections of the root (the root base) are exposed to plaque accumulation for more extended periods, thus fostering a more pronounced formation of IP [59].

A wide variety of plants were proven to form IP, comprising underwater plants, emergent plants, terrestrial plants in aquatic environments, and so on. Representatives of these plants are *Oryza sativa*, *Camellia sinensis*, *Iris pseudacorus*, *Canna indica*, *Rhizophoraceae*, *Acorus gramineus* L., *Jumex bulbosus*, *Pistia stratiotes* L. and *Elodea canadensis* [17,18,43,64]. The occurrence of IP is a naturally spontaneous phenomenon in the environment, influenced by a number of abiotic and biotic factors, such as soil properties, moisture levels, root oxygenation capacity, and so on (Figure 2) [65,66].

#### 2.3.1. Effect of Abiotic Factors on the Formation of IP

##### Soil Properties

The physiochemical properties of soils, including texture, organic matter (OM), pH, reduction potential (Eh) and elemental composition, exert significant and diverse influences on the formation of IP. Specifically, soil texture can influence IP formation by altering plant root growth, soil porosity, and the transport of rhizosphere elements. Notably, soils with lower clay content tend to favor the deposition of kinin on root surfaces compared to soils with higher clay content, as demonstrated by Chen et al. (1980) [46].

OM is an important soil component, comprising a wide variety of multifunctional groups derived from the decomposition residues of plants and animals. The degradation of OM facilitates the development of anaerobic or hypoxic conditions in the rhizosphere, hence creating favorable circumstances for root-oriented IP formation. Moreover, OM exhibits a great impact on the adsorption and migration behaviors of metallic elements in soils [67]. For instance, OM can effectively chelate more iron from its biogeochemical cycle, causing its accumulation at a higher level in the form of Fe(II) within the rhizosphere to be translocated in flooded environments [67].

The pH of soils significantly affects the concentrations of soluble Fe(II) and Mn(II) in soils, which are indispensable ingredients for IP aggregation via oxidation [62,68,69]. Under acidic circumstances, substantial amounts of iron and manganese are present in the form of soluble Fe(II) and Mn(II) in soils, which are favorable to IP formation, whereas, under alkaline circumstances, iron and manganese primarily exist as metallic hydroxides, which are not in favor of their further oxidation.

The Eh value takes charge of Fe(II) concentrations within the rhizosphere by influencing the diffusion rate of iron in soils [70]. Christensen et al. (1998) [71] found that in the soil sediments with appropriate Eh values, reduced forms of iron and manganese readily diffused toward the root surface. Then, the oxygen released by the roots triggered the oxidation of these elements into oxides on the root surface. Yang et al. (2012) [72] reported that in the soils adjacent to the hydrophyte rhizosphere, the Eh value was less than +50 mV, which was insufficient to gather enough Fe(II) ions to be oxidized within the rhizosphere. By contrast, Masscheleyn et al. (1991) [73] suggested that oxygen released from the roots could increase Eh values to at least +100 mV within the rhizosphere, benefiting a densified Fe(II) concentration for the further development of IP on the roots.

Non-metallic elements in the rhizosphere also have an effect on IP formation. Selenium (Se) and arsenic (As) are ubiquitous metalloids in natural soils, usually occurring with other metals in the form of oxides (e.g., selenite, SeO_3_^2−^ and arsenite, AsO_3_^3−^) [74]. Se could largely induce the response to oxidative stress in plants, resulting in an increase in the concentration of ROS in the roots and further stimulating the development of IP to inhibit the uptake of HMs [28,35,75]. Also, Se^0^ was found to facilitate the process of ROL in rice tissues, which was beneficial to advance IP formation on the roots [76]. Similarly, previous studies indicated that the exposure of As to the roots of hydrophilic plants advanced the development of IP, also by responding to the oxidative stress to generate a large amount of ROS, such as H_2_O_2_ and O^2−^ [35,75]. In addition, phosphorus (P) not only stands as a critical nutrient for plant growth but also impacts the biogeochemical cycles of iron in soil ecosystems [77,78,79,80]. The bioavailability of phosphorus in soils can affect the microbial community structure and activity, which play an important role in iron oxidation [81,82]. Furthermore, the addition of sulfide compounds in soils, e.g., hydrogen sulfide (H_2_S), can increase the root oxidation capacity of plants [59]. Rice plants grown in both soil pot and hydroponic settings supplemented with H_2_S, ranging from 2.64 to 5.28 mM, promoted the formation of IP Also, this concentration range of H_2_S enhanced rice growth, including seedling vigor, root length and the dry weights of roots and shoots [83,84]. However, it should be noted that in freshwater sediments, the excessive amount of H_2_S may not favor IP development on plant roots because H_2_S would chemically precipitate iron into the insoluble form of FeS and FeS_2_ [85,86].

##### Irrigation Regime

The different irrigation regimes give rise to different consequences concerning the synthesis of IP on root surfaces [87]. Excessively waterlogged soils will hinder the formation of IP. For example, continuous flooding has proven to significantly increase the population of Fe-reducing bacteria (FeRB), thus accelerating the reduction reaction of iron oxides around the root surface [21]. Therefore, a rational irrigation regime is important for creating a favorable environment for the accumulation of IP on the root surface [87]. Commonly, periodic flooding regimes can facilitate the conversion of IP from the amorphous form to the crystalline one [88,89,90,91]. This is because, in comparison to continuous flooding, periodic flooding (intermittent wetting and drying cycles) speeds up water movement in soils, stimulating bacterial growth and increasing the oxygen bioavailability [92,93]. Hence, under periodic flooding conditions, the amorphous iron was more likely to be transformed into the crystalline one within the rhizosphere soils with the abundant Fe-oxidizing bacteria (FeOB) and oxygen sources, thereby promoting the crystalline ratio of IP [94,95]. 

##### Other Factors

It is crucial to highlight the significance of planting density in maintaining soil health, recognizing that it is merely one of the numerous influencing factors. When considering planting density, it is evident that it plays a pivotal role in shaping the soil’s condition. Similarly, like other influencing factors, plant density has the potential to influence the Eh level of the soil and facilitate the formation of IP. This occurs because the roots of plants release additional oxygen into the soil when they are planted closer together. This extra oxygen not only enhances soil quality but also encourages the formation of IP. Therefore, when aiming to promote soil health and IP formation, it is imperative to consider planting density alongside other influencing factors, as supported by Christensen et al. (1998) [71] and Tripathi et al. (2014) [96].

The formation of IP is greatly augmented when exposed to combined HMs stress, rather than individual HMs stress. A recent study, conducted by Shen et al. (2021) [97], demonstrated that the presence of multiple HMs significantly enhances IP formation at the apical, middle, and basal regions of the root, with a gradual increase in formation over time. When mangrove plants are subjected to combined stress from Cu, Pb and Zn, they exhibit increased metal tolerance, which is associated with the substantial thickening and increased lignification and suberization of the exodermis. This enhanced lignification and suberization of the exodermis effectively delay the penetration of metals into the roots, thereby aiding in enhanced tolerance to heavy metals [98]. Furthermore, the deposition of increased lignin within the exodermis leads to a reduction in ROL emitted from mangrove roots [99]. The formation of IP on the root surfaces of mangrove plants is intricately intertwined with root ROL, as observed by [100] and Dai et al. (2017) [81].

Biochar, a carbon-rich and porous substrate, exhibits the ability to adsorb organic compounds and nutrients from the soil, thereby enhancing its fertility [101]. This adsorption capacity may influence the concentration of iron ions in the soil, potentially affecting the formation of IP. By fortifying biochar with iron (referred to as DCB-Fe), it significantly augments its specific surface area and enhances its surface functional groups, leading to an increase in its adsorptive capacity for heavy metals (HMs) [48,102]. The incorporation of biochar into soils is considered an environmentally friendly strategy to mitigate soil contamination, enhance phytoremediation, and reduce health-related hazards. The remediation efficiency of biochar in soils depends on various factors such as soil pH, HM content and porosity [15].

Biochar amendments were observed to increase pH and phosphorus levels in soil pore water, resulting in an elevation of IP formation on the root surface. This elevation was shown to reduce concentrations of Cd, Zn and Pb in rice shoots by up to 98%, 83% and 72%, respectively [103]. Similarly, laboratory experiments using nano-Fe_3_O_4_-modified biochar have demonstrated that its application promotes IP formation, thereby enhancing the root barrier against Cd [48].

Recent studies have shown that biochar derived from rice straw can be used to reduce Cd, Pb and Zn accumulation in rice shoots. However, it simultaneously increases As content. This increase in As content may be attributed to a decrease in soil pH, which promotes the conversion of As(V) to the more soluble and toxic As(III) form [103,104].

#### 2.3.2. Biotic Factors Effect on IP Formation

Apart from abiotic factors, it is generally acknowledged that numerous biological elements are capable of either directly or indirectly influencing the formation of IP. These influential factors span a wide spectrum, encompassing aspects such as microbial activities [76,105,106,107], durations of root oxidation [100], the presence and nature of root exudates [108], enzymatic activity within the plant [24], the genotypic variety of the plant [109], as well as the specific cultivar and age of the plant [110]. These factors collectively contribute to the ecological dynamics that govern IP formation on plant roots.

Expanding on this foundation, the species and genotypes of plants significantly influence the formation of IP, particularly through their impact on radial oxygen loss (ROL). Diverse hydrophytes, including *Typha latifolia* L., *Phragmites communis* L., and *Oryza sativa* L., exhibit varied capabilities in forming IP [111]. For instance, in rice, variations in IP formation among different genotypes and varieties are attributed to disparities in oxygen secretion capacity. These differences critically affect the plants’ ability to oxidize and precipitate iron around their roots, thereby influencing the extent and nature of IP formation [35,39,112,113]. This highlights how specific biological traits of plants can interact with their environment to modulate their physiological responses and adapt to varying conditions.

##### Radical Oxygen Loss (ROL) Facilitated by Aeration Tissues (Aerenchyma)

Aerenchyma, a plant tissue featuring thin walls and sufficient intercellular spaces, serves as the primary conduit for oxygen transport from above ground to below ground, which is generally classified into two types: schizogenous and lysigenous aerenchyma [63,114]. Schizogenous aerenchyma forms gas spaces through cell separation and differential cell expansion, while lysigenous aerenchyma results from the death and lysis of specific cells in cereal crops like rice [115], maize [116], wheat [117] and barley [118].

Aerenchyma plays a crucial role in the growth of wetland and IP formation due to its interconnected intercellular spaces, which form an efficient ventilation system facilitating gas exchange [119]. This system enables the transfer of oxygen produced during photosynthesis to the roots, while also providing buoyancy and structural support to the plant [120]. For instance, Hydrophytes, such as rice, utilize aerenchyma to transport captured oxygen to the roots for metabolic activities and distribute the remaining oxygen to the entire rhizosphere through pressurized ventilation or simple diffusion [28,121]. Beyond that, species like *Cyperus alternifolius* L., subsp. *flabelliformis*, *Myriophyllum spicatum* L., *Vallisneria spiralis* L., and *Juncus effusus* L. develop aerenchyma to preserve air and release oxygen from their roots into the rhizosphere. This process leads to the transformation of hazardous dissolved substances into less toxic, insoluble, or unabsorbed forms (Fe^3+^, FeOOH, Mn^3+^, NO^3−^) [122,123].

Under anaerobic conditions, aerenchyma provides a diffusion pathway that reduces the resistance of oxygen transport from the plant’s above-ground parts to the flooded or oxygen-deficient roots, ensuring the metabolic needs of the roots and contributing to ROL [76,124,125,126].

ROL stands as one of the most pivotal processes that trigger the formation of IP and oxidized root channels [79,112]. ROL was demonstrated to exert a substantial influence on the pH, Eh and the balance between Fe(II) and Fe(III) in the rhizosphere [127]. Through ROL, plants can effectively release or diffuse oxygen into the rhizosphere [128]. Consequently, Fe(II) readily undergoes oxidation to Fe(III) and precipitates onto the root surface in the form of hydroxide or hydroxyl oxide, thus giving rise to Fe plaque [20,99,129,130,131]. ROL is regulated by oxidation-reduction reactions mediated by ROS, and different wetland plant species exhibit varying root porosity and ROL rates [13,132]. Research has shown that rice genotypes with higher ROL rates have a more pronounced impact on pH, Eh and the balance between Fe(II) and Fe(III) in the rhizosphere. This results in the formation of a more extensive Fe plaque on the root surfaces compared to genotypes with lower ROL rates [127]. This underscores the significant role of ROL in plaque formation. Furthermore, Bravin et al. (2008) [133] established that the ROL capacity of rice roots and the soil’s buffering capacity are crucial factors affecting oxidation-reduction changes in the rhizosphere.

##### Hydrophyte Oxidative Systems

Hydrophyte roots possess a robust oxidation system capable of oxidizing metal ions present in the environment. This system, owing to its ability to form IP, protects the root zone from harmful substances. The oxidation system comprises root exudates and enzymatic activities of plant root [134]. They both reduce the Fe(III) into soluble Fe(II) in the rhizosphere, preparing Fe(II) for IP formation [135].

Root exudates are essential components of the oxidative secretions released by plants, playing a critical role in the transformation and mobility of Fe and Mn [136]. Root exudates (organic acid, phytosiderophores, etc.) were also documented to mitigate HM toxicity, including Al, Zn and Cd, through the exudation of glyoxylic, oxalic, and formic acid [137,138,139,140,141,142]. For instance, the oxalate content in the roots was observed to increase upon treatment with Pb in Pb-resistant rice varieties, as demonstrated by Yang et al. (2000) [143], highlighting its potential for HM blocking. Furthermore, excessive organic acids can be enzymatically decomposed into harmless CO_2_ and H_2_O_2_, as reported by Ando et al. (1983) [144] and Emerson et al. (1999) [105].

Enzymatic activities play an important role in oxidating Fe(II) [145]. The enzymatic antioxidant system maintains a delicate balance between the production and removal of ROS. ROS, comprising superoxide anion (O^2−^), singlet oxygen (1O^2^), hydrogen peroxide (H_2_O_2_) and hydroxyl radical (OH), are prevalent in plant cells [146]. They play a vital role in cellular metabolism and signal transduction. However, the excessive ROS production in plants causes oxidative stress and damage to biological molecules under stress (HMs, slat or abnormal temperature), leading to cellular dysfunction or death [147]. In this case, the activity of the enzymatic antioxidant system is accelerated. The superoxide dismutase (SOD) and catalase (CAT) activities involved in the elimination of ROS are activated [148], resulting in a large amount of O_2_, which is beneficial to create an oxidizing environment in the rhizosphere [147].

##### Fe-Reducing and Fe-Oxidizing Bacteria

The oxidation of iron can occur through two distinct pathways in nature: chemically driven and biologically driven oxidation, which is decided by the concentration of oxygen. The former is primarily driven by chemical catalysts (≥275 μM), while the latter, is dominated by microbial activity (≤50 μM) [147,149,150,151,152]. Therefore, in an anaerobic or anoxic environment, biologically driven iron oxidation predominates in IP formation. Among all the microbes, iron-oxidizing bacteria (FeOB) and FeRB serve as the primary driving force in the vicinity of wetland plant roots. 

FeOB can be classified into four types [152], namely acidophilic aerobic, neutrophilic microaerobic, anaerobic phototrophic, and nitrate-reducing [153], which significantly impact the kinetics of Fe(II) oxidation and oxygen consumption at the anoxic interface around the roots [154]. The acidophilic aerobic and neutrophilic microaerobic Fe(II)-oxidizers contribute the most during the IP formation [155]. Two distinct types of FeOB synergistically function in wetland and flooded environments, imparting numerous advantageous effects and contributing to the formation of IP under anaerobic conditions. Neutrophilic microaerobic Fe(II)-oxidizers were initially discovered by Ehrenberg in 1836 and subsequently purified and isolated in the 20th century [156]. These bacteria have since emerged as crucial model organisms for investigating Fe(II) oxidation and associated environmental processes [157]. They are commonly found in neutral environments, including soil, the aerobic–anoxic interface of redox-stratified aquatic systems, plant rhizospheres, groundwater flow zones and deep-sea sediments. Under these neutral microaerophilic or anaerobic conditions, they utilize Fe(II) as an electron donor and O_2_ as an electron acceptor, while organic or inorganic carbon sources facilitate their growth [158]. Consequently, the precipitation of Fe(III) occurs on the root surface in the form of FeOOH along with other elements. Acidophilic iron-oxidizing bacteria was first isolated by Colmer and colleagues in 1947 [159]. These bacteria typically inhabit acidic environments with a pH range of 1.0–4.0 [160], such as acid leachate, acid mine drainage (AMD), deep-sea hydrothermal vents, and hot springs that are abundant in iron, sulfur, and other metallic elements. Within these acidic habitats, Fe(II) remains stable and bioavailable for microbial utilization, enabling iron-oxidizing microorganisms to outcompete oxygen-mediated abiotic oxidation processes for acquiring Fe(II). Consequently, they thrive by utilizing elemental S or Fe(II) as electron donors while employing O_2_, SO_4_^2−^, or NO_3_^−^ as electron acceptors [161]. Additionally, organic or inorganic carbon serves as their carbon source. In a study on reeds, the presence of acidophilic FeOB not only enhances the formation of IP but also diminishes the uptake of Fe and Mn by the reeds. It is postulated that FeOB facilitates IP formation in acidic environments, thereby indirectly impeding heavy metal absorption [162].

FeRB accounts for 12% of all rhizosphere bacteria and are dominant members in the rhizosphere microbial community, along with the FeOB [163]. They utilize hydrogen (H_2_) and acetic acid as electron donors to reduce Fe(III) to Fe(II) under anoxic conditions [164]. This makes IP an ideal electron acceptor for FeRB [165], decreasing iron precipitation by influencing both Fe reduction and Eh in soil [47]. Beyond that, the presence of FeRB in mangrove wetland sediments could potentially impact the phase transition of iron oxide. In a controlled climate chamber experiment conducted by Zhang et al. (2023) [166], it was observed that inoculation with FeRB strain *Pseudomonas sp. SCSWA09* significantly decreased IP formation on the roots of *Kandelia obovata* seedlings, particularly reducing amorphous IP. This reduction can be attributed to the ability of FeRB to expedite the transformation from amorphous ferrous/ferric hydroxide into crystalline forms, suggesting their influence on IP generation and implying a potential acceleration of active iron cycling in the rhizosphere.

It is important to note that the formation of IP is regulated by a complex biological system involving interactions between biotic and abiotic factors. For instance, secretions from wetland plant roots (such as glucose, glycine, citrate and malate) are oxidized by microorganisms into carbon dioxide, which can impact the pH of the rhizosphere [167]. Additionally, an oxidation reaction occurs outside the cell wall and produces protons, thereby influencing the pH of the soil [168]. Similarly, Johnson-Green and Crowder (1991) [169] reported significant differences in Fe solution pH after exposure to axenic and non-axenic seedlings. This suggests a weak trend of competition between iron-oxidizing bacteria and chemical oxidation of Fe(II) at low pH levels. Under such conditions, Fe oxidation kinetics are relatively slow (<4), but acidophilic FeOB like Thiobacillus ferrooxidans may enhance Fe oxidation kinetics and contribute to IP formation [105]. Therefore, there could be interactions among plants, environmental substances, and microbes during this process.

### 2.4. IP as an Armor for Metal Transfer in Plants

After decades of extensive research on IP, numerous significant discoveries were made regarding the presence of IP, which effectively enhances plant resistance against HM toxicity in soil. In the case of rice, Greipsson and Crowder (1992) [38] observed that exposure to 0.5 mg∙L^−1^ Cu(II), 2.0 mg∙L^−1^ Ni(II) and a combination of Cu(II)+Ni(II) resulted in chlorosis and necrosis in non-IP rice plants, whereas IP rice plants exhibited no signs of toxicity throughout their growth period. In rice exposed to excessive Zn and Cu [170], IP positively impacted the dry weight of shoots and roots, leaf and root length, and reduced the occurrence of chlorotic leaves when exposed to excessive Cu [96]. Moreover, under Cd stress, the concentration of Cd(II) in the root and bud as well as the transfer of Cd(II) from root to bud in rice with IP decreased by 34.1%, 36.0% and 20.1%, respectively, compared to rice without IP [17]. As is a highly toxic and carcinogenic metallic substance that can be readily absorbed by rice in significant quantities [171]. The presence of IP effectively inhibits As uptake by roots, thereby reducing its accumulation in brown rice [172]. More surprisingly, IP can also oxidize As(III) to the less toxic As(V), thus reducing the toxicity of As to plants [78,173]. It should be noted that when As is oxidized to arsenate by oxygen, IP will reduce absorption [96,113], which may be due to their different structure, resulting in different binding capabilities or ways to IP [174]. Li et al. (2016) [66] selected three distinct types of paddy soils, denoted as C, D and N, and artificially manipulated the effective concentrations of Pb by adding 0 mg/kg, 150 mg/kg and 300 mg/kg Pb(II) to soils C, D and N, respectively. Despite higher effective concentrations of Pb in soils D and N compared to soil C, rice plants exhibited significantly lower levels of Pb absorption in these soils. This phenomenon is attributed to a substantial presence of IP coating on the surface of rice roots which effectively reduces the mobilization of Pb in both soil types D and N. These findings suggest that IP generally acts as a protective barrier against toxic metals while enhancing plant growth [175].

Currently, numerous studies have been conducted on the mechanism of IP blocking the absorption of HMs by plants [176]. Chemically, most plant roots possess a negative charge, enabling them to adsorb positively charged HMs [177]. The presence of IP physically obstructs the interaction between roots and positively charged HMs [178,179]. Moreover, due to the abundant functional groups present in iron hydroxides, IP can effectively sequester metal(loid)s through adsorption and/or co-precipitation processes. Consequently, this may influence the availability of metal(loid)s in the rhizosphere and subsequently impact the uptake and accumulation of HMs by plants [180].

Physically, the adsorption mechanism of IP to HM is generally inferred by studying the natural minerals contained in IP Wang et al. (2009) [181] choose goethite, magnetite, e.g., (provided by Sinopharm Chemical Reagent Co., Ltd., Shanghai, China) as representatives of metal (hydr)oxides commonly present in nature, and then found that Cd was absorbed on these different oxide minerals. Hochella et al. (1989) [182] found that the surface structure and nano-scale morphology of minerals play a key role in the dissolution and adsorption reaction between the surface and the soils. The iron isotopic exchange experiments show that ferrihydrite contains labile and non-labile site populations; the number of sites participating in the faster exchange process was reduced by adsorbing arsenate before the exchange experiment. The labile sites, examined with Mossbauer spectroscopy, are found to have different local environments; compared to sites that exchange slower, sites that exchange very quickly (within 20 min) had more distorted octahedral geometry. When bonded to adsorbed arsenate, the distortion oflabile sites was slightly reduced. Adsorbed arsenate may decrease the degree of distortion around the octahedra by forming binuclear, bidentate bonds with the adjacent iron octahedra [183]. Arsenate adsorbs on ferrihydrite surfaces mainly as an inner-sphere bidentate (bridging) complex sharing apical oxygens of two adjacent edge-sharing Fe oxyhydroxyl octahedra. Monodentate complexes were also observed, accounting for about 30% of all As–Fe correlations [184]. Fuller et al. (1993) [185] analyzed kinetics of ferrihydrite to adsorp and coprecipite arsenate. In adsorption experiments, a period of rapid (5 min) As(V) uptake from solution was followed by continued uptake for at least eight days, until As(V) diffused to adsorption sites on fenihydrite surfaces within aggregates of colloidal particles. The time dependence of As(V) adsorption is well described by a general model for diffusion into a sphere if it was assumed that the subset of surface sites located near the exterior of aggregates can quickly reach the adsorption equilibrium. In coprecipitation experiments, the initial As(V) uptake was significantly greater than in post-synthesis adsorption experiments because As(V) was coordinated by the surface sites before the process of crystallite growth and aggregation; therefore, the absorption rate was not affected by diffusion. After the initial adsorption, As(V) was slowly released from coprecipitates for at least one month, because crystallite growth led to desorption of As(V). In addition, numerous adsorption models were extensively developed, such as the diffusion layer model [186], three-layer complexation model [187], modified three-layer complexation model [188] and the metal (hydroxide) oxide surface reaction group affinity valence band theory [189], these models proceed from different angles, explaining the adsorption behavior of iron oxide minerals.

### 2.5. Native Plants in Phytoremediation: Interactions and Ecological Effects on Soils and Plants in Heavy Metal Contaminated Environments

#### 2.5.1. Effect on Plants and Soils

IP, as a key product of the iron oxidation-reduction cycle, plays a significant role in transforming trace metals and organic matter in flooded soils, which contain high levels of iron ions [190].

The impact of IP on plant growth remains a hotly debated issue. Its influence varies depending on the heavy metal environment surrounding the plant. For instance, a study on water lobelia *(Lobelia dortmanna* L.) revealed that IP does not affect the root diameter [191]. In research involving common bulrush (*Typha latifolia*), it was observed that the presence or absence of IP had no significant effect on the dry weight of roots and shoots of seedlings, whether in control conditions or in Zn and Cd solutions. However, roots were significantly shorter when IP was present [33]. Similarly, Greipsson et al. (1994) [178] found that under Ni and Cu stress, rice roots were shorter when IP was present. Although IP does not always promote root growth, it does not necessarily imply a negative impact on overall plant growth. For example, Møller and Sand-Jensen’s study showed that IP around the roots of *Lobelia dortmanna* L. creates an oxygen diffusion barrier [192], which can be beneficial in high-sediment environments by directing more oxygen to root meristems, thereby improving survival. Likewise, while IP resulted in shorter rice roots under Ni and Cu stress, it significantly enhanced rice shoot growth [176]. This suggests that the formation of IP is an effective response of plants to various environmental stresses.

Moreover, IP influences the chemical behavior and bioavailability of nutrients [193,194,195], acting as a nutrient reservoir to store essential elements [196]. IP on roots serves as an iron reserve, aiding plants in overcoming iron deficiency. For instance, in Medicago sativa under Cd stress, IP formation enhances photosynthesis efficiency and biomass production [62]. IP enriches environmental phosphorus, thereby enhancing plant energy metabolism, nucleic acid biosynthesis, photosynthesis, enzyme activities, and the biogeochemical cycles of iron and manganese [80]. Additionally, when Fe(OH)_3_ is added to phosphorus-rich nutrient solutions, a significant increase in the P content of rice shoots is observed, correlating positively with the amount of IP attached to the roots [197]. The adsorption function of IP mainly stems from its primary component, Fe(OH)_3_. Its amphoteric colloid properties and loose, porous structure provide a large surface area, facilitating the absorption of phosphorus and other elements [196,198,199]. Furthermore, IP is suggested to contribute to increased nitrogen accumulation in tea plant roots and stimulate plasma membrane ATP enzyme activity [200].

IP also has an impact on the rhizosphere soil of plants. The formation and reductive dissolution of IP can significantly influence the rhizosphere’s iron budget, affecting the mobilization of soil pollutants and nutrients [130]. During its formation, there is a release of H^+^ and the secretion of various organic acids such as malic acid, lactic acid, oxalic acid, citric acid and succinic acid into the rhizosphere [60,201]. These processes lead to changes in the pH and Eh values of the rhizosphere soil, subsequently affecting the bioavailability and concentration of HMs [23,26,165].

IP also acts as a barrier against oxygen loss, enhancing oxygen supply to the root meristems. This, in turn, influences the composition and distribution of aerobic and anaerobic microorganisms in the soil [36]. For example, during the iron oxidation process, the relative abundance of copper bacteria such as Maxilla, Pseudomonas, Rosella, Coleopomonas and Proteus increases, eventually becoming dominant [202]. This suggests that the formation of IP leads to a more stable microbial community structure, aiding our understanding of the transformation of organic matter and HMs [203]. Figure 3 shows a sketch of the interaction between plants, IP and microbes.

#### 2.5.2. Ecological Role in Environmental Remediation

In today’s era of rapid industrial and agricultural technological advancement, environmental pollution, such as heavy metals, has become a global concern. Waste materials from industrial and agricultural activities are discharged into natural environments through sewage and sludge, eventually entering agricultural soils and posing risks to human and environmental safety [204,205,206,207].

Among various soil remediation methods, bioremediation is favored for its cost-effectiveness and eco-friendliness. For instance, *Pteris vittata* L., known for its robust growth and high tolerance to HM toxicity, is used for biomonitoring and assessment of metal pollution in sediments [208]. As a crucial link between plants, microorganisms and soil, IP plays a significant role in the remediation of soil heavy metals. Wetland plants use their aeration tissues to transfer oxygen to their roots, while the low redox potential of sediments leads to the gradual accumulation of substances like Fe(II), Mn(II), H_2_S and CH_4_ [38]. These conditions create an ideal environment for microbial survival, and both the oxygenating ability of roots and microbial activity support the formation of IP in aquatic plant roots [209]. The IP attached to the root surface provides a large binding surface area for the absorption of metals and other elements, effectively remediating polluted soils [36].

Numerous researches indicated that plants with IP adherence, which thrive with high biomass in saline water conditions and possess deep root systems, can flourish in challenging environments and demonstrate strong metal accumulation capabilities [210], especially in their roots [211]. For example, *Spartina alterniflora Loisel.* is identified as an effective species for remediation due to its ability to accumulate considerable amounts of certain HMs (Cd, Cr and Mn), and lead, in its above-ground parts [212]. Jia et al. (2018) [213] confirmed that the IP characteristics of wetland plants can regulate iron, manganese and phosphorus in agricultural drainage.

Apart from HMs, environmental pollutants that pose a threat to plant survival include persistent organic pollutants (POPs), microplastics (MPs) and emerging contaminants (ECs) such as waterborne antibiotics and sterol hormones [100]. Under waterlogged conditions, certain steroidal hormones and waterborne antibiotics can accumulate on IP, with high adsorption sites and functional groups of IP facilitating their removal [214]. Polycyclic aromatic hydrocarbons (PAHs) and polybrominated diphenyl ethers (PBDEs) are persistent organic pollutants commonly found in both industrial discharges and ecosystems [215]. IP can serve as a physical barrier to impede the entry of contaminants into plants, contributing to the immobilization of PAHs and PBDEs [216].

### 2.6. Perspectives and Conclusion

#### 2.6.1. Exploring the Cultivation of High-Yielding IP Plant–Microbe Combinations to Address Environmental Pollution

The formation of IP is a natural process that does not cause secondary pollution, making it a promising strategy for remediating soils contaminated with HMs. The plant–microbe–soil system is complex, and a more comprehensive study is needed to understand the interactions within polluted soils. Meng et al. (2024) [217] revealed for the first time the process of IP generating highly reactive hydroxyl radicals (·OH) and verified the role of the produced ·OH in the oxidation and transformation of pollutants in the rhizosphere. In addition to pollutants, the produced ·OH may also affect the redox cycling of elements and the composition of the rhizosphere microbial community [218], subsequently impacting the growth of rice. Correspondingly, rhizospheric FeOB and FeRB may significantly alter the composition of IP, thus affecting the generation of ·OH by IP. In the future, the oxidative effects induced by ·OH generated from IP should be incorporated into the framework of understanding IP’s impact on rice plants.

#### 2.6.2. Policy and Sustainability

Integrating IP research into environmental protection policy involves developing guidelines and strategies that utilize IP to mitigate soil and water pollution. Governments and environmental agencies can formulate policies that encourage the use of plants with high IP-forming capabilities in areas affected by heavy metal contamination. This approach can be part of a broader environmental restoration plan aimed at reducing the impact of industrial and agricultural pollutants on ecosystems. In addition, policies can provide funding and incentives for research into the mechanisms of IP formation, its impact on plant growth and soil health and methods to enhance IP formation in different plant species. Such research could lead to new agricultural practices and environmental remediation techniques.

## 3. Conclusions

Plants have evolved adaptive and versatile strategies to perceive and respond to fluctuations in element availability, optimizing their growth, development and reproduction under changing environmental conditions. These strategies encompass a range of mechanisms, from chelation and osmoregulation to antioxidant systems, including root secretions, cell walls, cell membranes and vacuolar compartmentalization. These factors significantly influence the mobility of heavy metals (HMs) and microbial activity [219,220,221]. Increasingly, IP is being recognized as a microbial armor and nutrient treasury for plants.

The intricate plant–microbe–soil system poses significant challenges in comprehending the interactions within polluted soils. FeOB plays a pivotal role in the iron cycle, yet the metabolic pathways of these bacteria and their involvement in the iron cycle around roots remain enigmatic. It is imperative to investigate the ecological and environmental implications of IP to gain a deeper understanding of its significance.

In conclusion, a profound comprehension of the intricate interactions among plants, rhizosphere microorganisms and polluted soils is crucial for addressing the environmental impacts of soil pollution and developing effective remediation strategies. Key avenues for future research include exploring the metabolic pathways of FeOB and assessing the safety and practical significance of IP plants.

## Figures and Tables

**Figure 1 plants-13-01476-f001:**
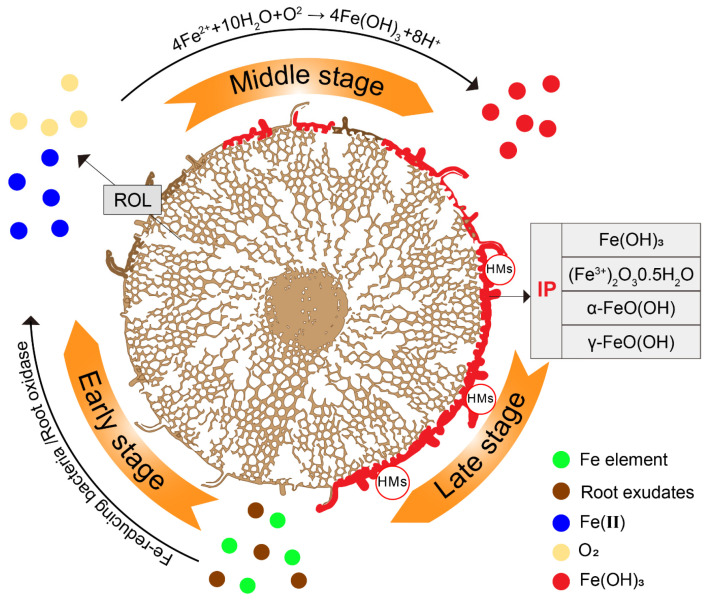
The formation process of IP. Through various oxidation-reduction processes occurring outside the root (shown as cross-section), a large amount of soluble Fe(II) is formed, which is readily oxidized by dissolved oxygen in soils. According to the equation of 4Fe(II) + 10H_2_O + O_2_→ 4Fe (OH)_3_ + 8H^+^, this results in the rapid precipitation of iron oxide on the surface of roots.

**Figure 2 plants-13-01476-f002:**
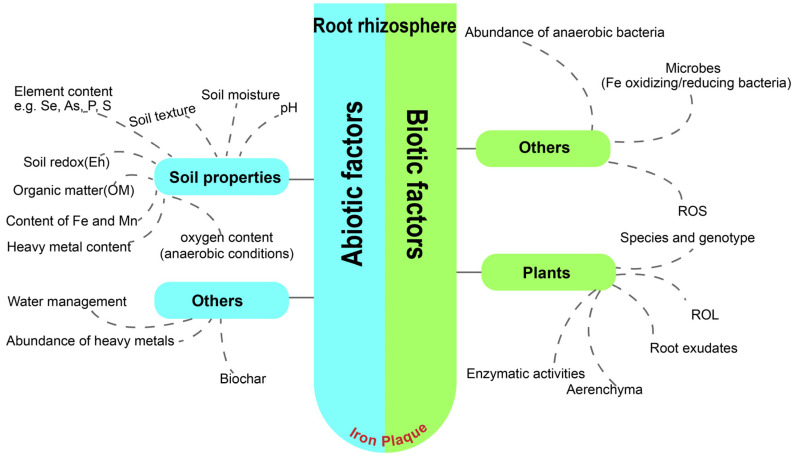
Schematic diagram of abiotic and biotic factors influencing the IP formation in the rhizosphere.

**Figure 3 plants-13-01476-f003:**
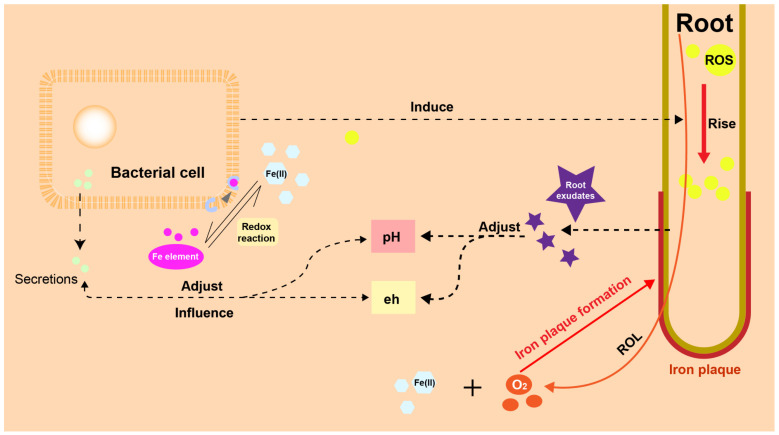
Interaction between plants, IP and microbes. In the rhizosphere environment, plants and microorganisms have their own strategy to influence pH and eh of soil. Meanwhile, microbes can influence plants by rising ROS in root cell.

**Table 1 plants-13-01476-t001:** The species of HMs that were reported to be blocked by IP.

Plants.	HMs	IP Composition	Reference
*Typha latifolia*	As	Ferrihydrite (Fe_2_O_3_.nH_2_O) Lepidocrocite (γ-FeOOH) Goethite (α-FeOOH) Siderite (FeCO_3_)	Hansel et al. (2002) [31]
*Phalaris arundinacea*	As	Ferrihydrite (Fe_2_O_3_.nH_2_O) Goethite (α-FeOOH) Siderite (FeCO_3_)	Hansel et al. (2002) [31]
*Phalaris arundinacea*	Mn, Pb, Zn	Ferrihydrite (Fe_2_O_3_.nH_2_O) Goethite (α-FeOOH) Siderite (FeCO_3_)	Hansel et al. (2001) [32]
*Typha latifolia*	Zn, Pb, Cd	Not mentioned	Ye et al. (1998) [33]
*Oryza sativa* L.	As	Not mentioned	Deng et al. (2010) [34]; Lee et al. (2013) [35]; Xiao et al. (2020) [36];
*Oryza sativa* L.	Cd	Not mentioned	Liu et al. (2007) [37]
*Oryza sativa* L.	As	Not mentioned	Deng et al. (2010) [34]; Lee et al. (2013) [35]; Xiao et al. (2020) [36];
*Oryza sativa* L.	Cd	Not mentioned	Liu et al. (2007) [37]
*Oryza sativa* L.	Cu, Ni	Not mentioned	Greipsson and Crowder (1992) [38]
*Oryza sativa* L.	Cr	Not mentioned	Zandi et al. (2020) [39]; Xu et al. (2018) [40]; Xiao et al. (2021) [41]
*Oryza sativa* L.	Zn, Cd	Not mentioned	Xu and Yu, (2013) [42]
*Pistia stratiotes* L.	Cd	Not mentioned	Singha et al. (2019) [43]
*Iris pseudacorus*	Cd	Not mentioned	Ma et al. (2020) [18]
*Spartina alterniflora*	Cu, Zn, Pb, Cr	Not mentioned	Zhang et al. (2020) [44]; Xu et al. (2018) [45]

**Table 2 plants-13-01476-t002:** The iron oxides and hydroxides [49].

Oxides	Hydroxides and Oxide-Hydroxides
Hematite α-Fe_2_O_3_	Ferrihydrite Fe_2_O_3_·nH_2_O
β-Fe_2_O_3_	Goethite α-FeOOH
Maghemite γ-Fe_2_O_3_	Lepidocrocite γ-FeOOH
ε-Fe_2_O_3_	Bernalite Fe(OH)_3_
Magnetite Fe_3_O_4_ (Fe^2+^${Fe}_{2}^{3+}$O_4_)	Akaganéite β-FeOOH
Wüstite FeO	δ-FeOOH
	Feroxyhyte δ′-FeOOH
	High pressure FeOOH (laboratory compound)
	Fe(OH)_2_
	Schwertmannite Fe_16_O_16_(OH)_y_(SO_4_)_z_·nH_2_O
	Green rusts: ${Fe}_{x}^{3+}{Fe}_{y}^{2+}$(OH)_3x+2y−z_($A^{-}$)_z_; $A^{-}$= ${Cl}^{-}$;1/2S$O_{4}^{2-}$

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
