# Peer review of "Iron Plaque: A Shield against Soil Contamination and Key to Sustainable Agriculture"

_plants, 2024, doi:10.3390/plants13111476_

Round 1
Reviewer 1 Report
Comments and Suggestions for Authors
Iron Plaque: A Shield Against Soil Contamination and Key to
Sustainable Agriculture : A review.
The authors have presented a detailed review of the factors related to the management and protection of crops in flooded crop land. The cited articles report the complex interactions of chemical, microbiological, and crop biochemistry. Some suggestions have been added for new research.
This review should be published as a source of information for the planning and evaluation of new research on the complex interactions of crops in flooded soils.
Reviewer 2 Report
Comments and Suggestions for Authors
The review focuses on various aspects concerning the formation and functioning of iron plaque on plant roots. This is an important and timely topic which fits the scope of the Journal. The review is well-structured. It covers the discovery and chemical composition of iron plaque, its formation and functioning, the effects of abiotic and biotic factors on iron plaque formation, the role of iron plaque in protecting plant from heavy metal effects as well as future prospects in the field. The tables and figures are informative. The citations are relevant.
There are some minor remarks:
«…crystallinity of IP occurred on the roots of Spartina alterniflora increased as Fe(II) concentration in soil…» - It is a bit unclear. Maybe “increased with Fe(II) concentration in soil”?
Is it known whether IP in the root elongation zone can affect the rate of root elongation? If so, this could result in the formation of shorter roots.
Figure 2: Maybe it would be more logical to move “species and genotype” to the “Plants” section instead of “Other”.
“increased metal tolerance, which is associated with the delayed function of the lignified/suberized exodermis. This delayed function acts as a barrier, preventing the entry of HMs into the roots (Cheng et al., 2014).” - It is unclear what is meant by “delayed function”. Please rephrase and explain.
“Root exudates, a crucial constituent of the oxidative secretions released by plant, which strictly relevant to transformation and mobility of Fe and Mn” – this is unclear, please rephrase.
“Enzymatic activities is decisively in oxidating Fe(II).” – This is unclear. Please check grammar and revise.
“which decided by the concentration of oxygen” – This is unclear, please check grammar and rephrase.
“biologically-driven iron oxidation is prevailed IP formation.”- Please check grammar and revise.
“oxidizing bacteria (FeOB) and FeRB serves as the primary driving force” – please correct “serves” for “serve”
“The acidophilic aerobic and neutrophilic microaerobic Fe(II)-oxidizers contributes the most…” – please correct “contributes” for “contribute”.
“Wang et al. (2009) choose goethite, magnetite e.g.” - Please check grammar and revise.
Please correct “oc- tahedra.”
“many adsorption models have been proposed at home and abroad” – This is not clear, please rephrase
“ecological impact of indigenous phytoremediators” – This is not clear, please rephrase.
“this cycle in flooded soils, which are rich in highly iron ions,” – This is not clear, please rephrase.
“IP also enriches environmental P in the IP,” – This is not clear, please rephrase.
“biomonitory” should be “biomonitoring”
“high saltwater biomass” - This is not clear, please rephrase.
Page 11 – “O2, SO4-2, or NO3-” should be O2, SO42-, NO3- (please correct the indexes).
As this is a review article, the subheading “2. Results and discussion” doesn`t seem necessary.
The references in the text of the manuscript should be substituted with numbers according to the rules of the Journal, and the reference list should be formatted accordingly.
Please check that all Latin species names are given in italics (see p 9 - Typha latifolia L., Phragmites communis L., and Oryza sativa L.).
Comments on the Quality of English LanguageModerate editing of English language is required.
